# Structural, Microstructural and Magnetic Properties of SmCo₅/20wt%Fe Magnetic Nanocomposites Produced by Mechanical Milling in the Presence of CaO

**Razvan Hirian** [1], **Rares Adrian Bortnic** [1], **Florin Popa** [2], **Gabriela Souca** [1], **Olivier Isnard** [3,4] **and Viorel Pop** [1,*]

1   Faculty of Physics, Babeş-Bolyai University, Cluj-Napoca RO-400084, Romania
2   Materials Science and Engineering Department, Technical University of Cluj-Napoca, Cluj-Napoca RO-400641, Romania
3   Institut NÉEL, University Grenoble Alpes, 25 rue des martyrs, F-38042 Grenoble, France
4   CNRS, Institut NÉEL, 25 rue des martyrs, F-38042 Grenoble, France
*   Correspondence: viorel.pop@ubbcluj.ro

**Abstract:** In this work, we demonstrate the possibility of using a soluble ceramic material, 5 wt% CaO, as an additive for an SmCo₅/20wt%Fe exchange-coupled nanocomposite obtained by mechanical milling in order to inhibit the grain growth of the soft magnetic phase during annealing, which results in a more stable microstructure and an implicit improvement in the hard–soft interphase exchange coupling. Moreover, we show that the additive improves the phase stability of the composite material, reducing the amount of Sm₂Co₁₇-type phases formed during the synthesis process, an important aspect because Sm₂Co₁₇ is detrimental to the magnetic performance of the SmCo₅/20%Fe nanocomposite. These effects are reflected in a nearly 13% increase in the coercive field ($H_c$) and a 20% increase in the energy product, $(BH)_{max}$, for the powders produced using CaO as compared to pure SmCo₅/20%Fe nanocomposites processed in the same manner.

**Keywords:** mechanical milling; hard–soft exchange-coupled nanocomposites; spring magnet; CaO matrix processing

## 1. Introduction

High-performance permanent magnets (HPPMs), are critical components for a wide variety of applications, from mobile phones and computers to wind turbines and electric vehicles [1–3]. HPPMs are useful in advanced technologies due to their large energy product, $(BH)_{max}$. However, these materials are currently obtained using rare earth elements (Nd, Sm, Dy, etc.). This is a problem because rare earth elements are associated with significant geopolitical and environmental costs [1,2]. Additionally, current predictions show that the situation is only going to become more tenuous, in large part due to the growing demand for electric vehicles [4]. Currently, scientists are attempting to solve some of these issues by (i) increasing the energy product of current HPPMs [5], (ii) identifying new magnetic materials with low or no rare-earth content [6,7] (which can replace HPPMs in certain less demanding applications [2]) or (iii) developing new nanostructured materials that promise larger energy products ($(BH)_{max}$ up to 1 MJ/m³), reducing the rare earth content [8]. We discuss the latter approach in this work, namely exchange-coupled nanocomposites [9], which are nanostructured magnetic materials in which the high magnetization of a soft magnetic phase is stiffened by the high anisotropy of a hard magnetic phase. However, in order for stiffening to occur, the soft magnetic phase inclusion cannot exceed twice the domain wall width of the hard magnetic phase [9]. Many synthesis avenues have been attempted to date, with the best result obtained in thin films [10], $(BH)_{max}$ = 400 kJ/m³. However, due to the precise nanostructure control required, the synthesis of these materials (especially in powders and bulk) has proven difficult. The greatest challenge is maintaining

the microstructure required for a sufficient degree of interphase exchange [11–23]. For example, although a powder material with good magnetic properties can be produced, the consolidation of such powders through hot pressing or spark plasma sintering is likely to destroy the delicate microstructure (in large part due to grain growth). The novelty of the present work is the challenge of demonstrating the possibility of using the matrix milling method [15,16], i.e., the addition of a soluble ceramic material, 5 wt% CaO, to the $SmCo_5+20wt\%Fe$ exchange-coupled nanocomposite in order to inhibit the grain growth of the hard and soft magnetic phases. This reduction is very useful because in order to obtain an effective hard–soft interphase exchange coupling, the size of the soft phase inclusion must not exceed twice the magnetic domain wall width of the hard magnetic phase—in this case, $SmCo_5$. Moreover, the reduction in grain growth rate should allow for an improved thermal stability of the microstructure of the exchange-coupled nanocomposite, i.e., maintaining the critical dimensions for effective interphase exchange coupling, which is essential when applying (high-temperature) processing and consolidation methods, such as annealing and sintering.

## 2. Results and Discussion

The XRD patterns for $SmCo_5/20\%Fe$ annealed at 600 °C for 1 h, along with the patterns for $SmCo_5/20\%Fe+5\%CaO$ annealed at 600 °C for 0.5 h and 1 h, are presented in Figure 1. The pattern for $SmCo_5/20\%Fe$ annealed for 0.5 h can be found in the literature [21]. These measurements show that all of the samples have wide Bragg peaks, which are indicative of small crystallite sizes. Moreover, all samples are made up of a mixture of $SmCo_5$, $Sm_2Co_{17}$ (where some of the Co sites are occupied by Fe) and $\alpha$-Fe [22,23]; the CaO could not be identified, probably due to a combination of the limited remaining quantity of CaO and because this phase has elements with low X-ray scattering factors.

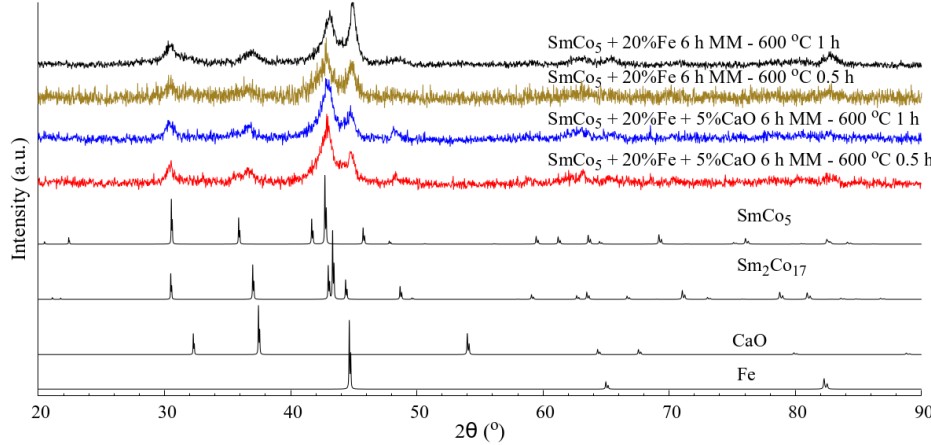

**Figure 1.** XRD patterns for $SmCo_5/20\%Fe$ and $SmCo_5/20\%Fe+5\%CaO$ exchange-coupled nanocomposite samples annealed at 600 °C for up to 1 h. The pattern for $SmCo_5/20\%Fe$ annealed for 0.5 h is taken from [21]. The calculated patterns for $SmCo_5$, Fe, $Sm_2Co_{17}$ and CaO are also presented for reference.

The most striking difference between the four nanocomposite materials is the intensity of the Fe peak (an approximately 2θ value of 45°), which varies drastically in the two investigated cases (with and without CaO addition). The $\alpha$-Fe peak in the case of the $SmCo_5/Fe$ nanocomposite material is always more intense in comparison to that of the samples produced with CaO. This difference may be due to the fact that Fe recrystallizes more readily in the absence of CaO.

Although the use of surfactants can produce oxidation [24], various oxides (iron oxide or samarium oxide) are absent due to the addition of CaO. This behavior can be explained by the very strong bonding between Ca and O. The affinity of O for Ca is high enough that it was used to reduce rare earth oxides into metals [15,16,25].

The difference is most striking in the case of the samples annealed for 1 h. The aforementioned Fe Bragg peak in the $SmCo_5$/Fe nanocomposite is nearly twice as intense as that in the nanocomposites with CaO addition. On the other hand, despite a considerable difference in intensity, the difference in the broadness of the Fe peak for the nanocomposites produced with and without CaO addition is minimal. This is reflected in the mean crystallite size (Table 1); namely the Fe crystallites in the $SmCo_5$/Fe sample annealed for 1 h are only 3 nm larger than those of the equivalent material produced with CaO addition. This could indicate that the Fe in the samples produced with CaO is very small and diffuse. Therefore not all of the Fe is visible in the diffraction pattern for the CaO sample, whereas in the case of $SmCo_5$/Fe sample, it is possible that the Fe recrystallizes into still small but noticeable crystallites.

**Table 1.** Crystallite sizes and phase composition for $SmCo_5$/Fe nanocomposites as estimated by XRD pattern analysis.

| Sample | D (nm) | $SmCo_5$ (wt%) | $Sm_2(Fe/Co)_{17}$ (wt%) | Fe (wt%) |
|---|---|---|---|---|
| 600 °C 0.5 h | $14 \pm 2$ | 60 | 26 | 17 |
| 600 °C 1 h | $17 \pm 2$ | 26 | 60 | 14 |
| CaO 600 °C 0.5 h | $14 \pm 2$ | 60 | 26 | 17 |
| CaO 600 °C 1 h | $14 \pm 2$ | 50 | 34 | 16 |

The approximate phase composition of the samples is also presented in Table 1. Because the signal-to-noise ratio of the XRD patterns is low and as a result of the high degree of convolution between the investigated phases, these values should only be taken as qualitative. However, the trend is clear; all the samples contain some $Sm_2Co_{17}$-type phase, which increase in quantity with annealing time (as the 1:5 phase only forms with Co, among the elements present in our samples, in order to form 2:17, some Fe must enter the structure of the 1:5 phase). However, the formation of the 2:17 phase is clearly inhibited by CaO addition. Both samples, with and without CaO addition, present with similar phase ratios; when annealed for 0.5 h at 600 °C, they contain approximately 26% the 2:17 phase. When annealing time is increased to 1 h, for the $SmCo_5$/Fe sample, the 2:17 content more than doubles to 60%, whereas in the case of the sample made with CaO addition, the 2:17 content only increases by approximately one-third. As proposed above, the CaO inhibits the diffusion of Fe within the material and also inhibits the growth of the 2:17-type phase, the growth of which is dependent on Fe diffusion and is, in general, detrimental to the performance of the exchange-coupled nanocomposite [21].

The SEM images for the produced nanocomposites are presented in Figure 2. At a scale of 50 μm, the particles sizes for the materials made with CaO addition are below 50 μm, with most particles being approximately 10 μm in diameter, irrespective of annealing time (Figure 2a,c). On the other hand, in the case of the $SmCo_5$/20%Fe nanocomposites, although some very small flakes can be observed, many of the particles are fused in large, seemingly compact structures (Figure 2e). This assessment is consistent with what was observed when recovering the samples from the milling vials; whereas the samples containing CaO were easily removed from the vial walls, removing the samples made without CaO addition took considerable effort, with samples coming out as a very coarse powder. At higher magnification, in all cases, the particles are observed to be made up of smaller micro- and nanoflakes that have fused together during the milling process (Figure 2b, 2d and 2f); this is typical of milled powders. Even the larger particles observed in the $SmCo_5$/Fe sample seem to be made up of the same small, fused flakes (Figure 2f).

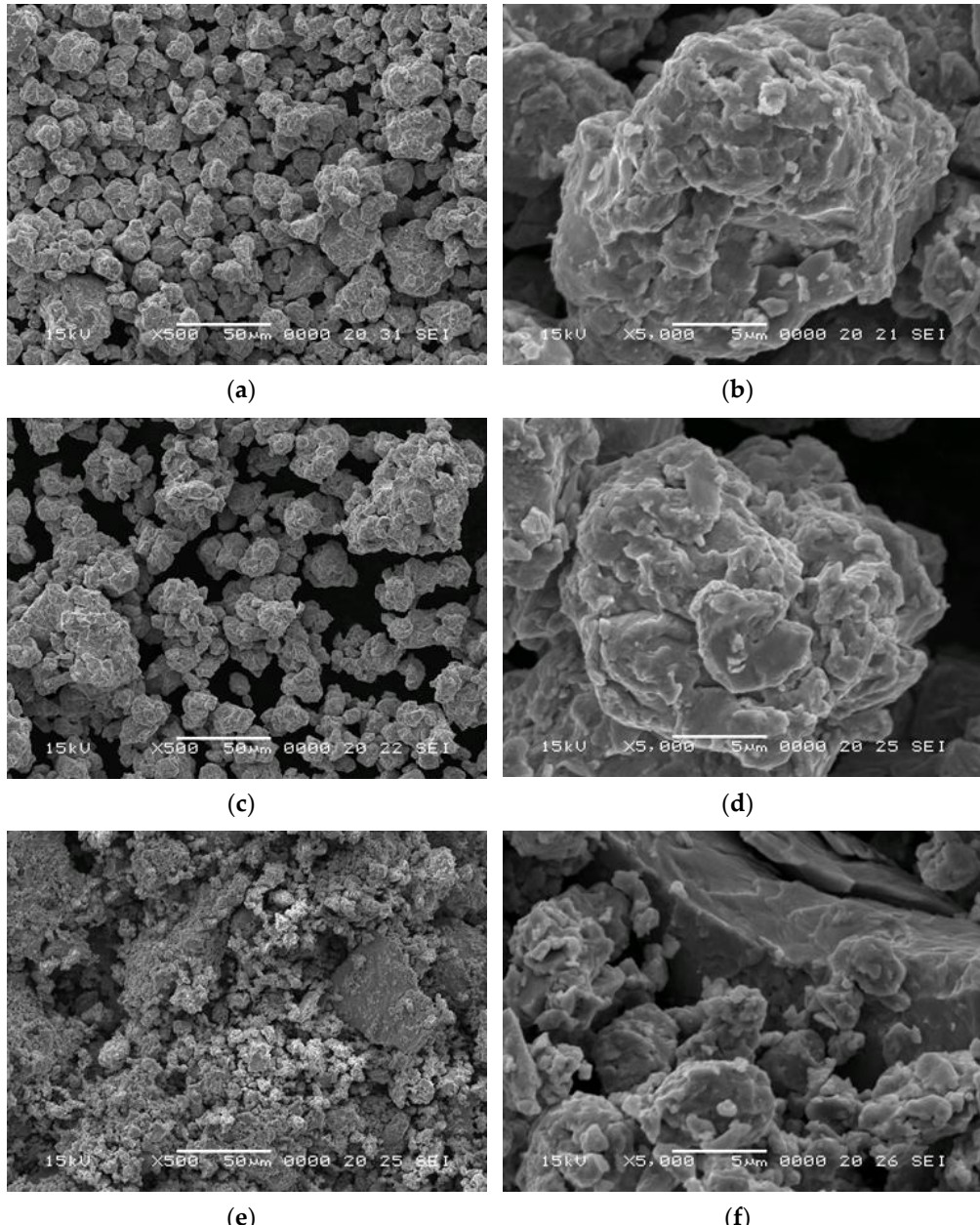

**Figure 2.** SEM images of $SmCo_5/20\%Fe+5\%CaO$ annealed at 600 °C for 0.5 h (figures (**a**) and (**b**), respectively) and 1 (figures (**c**) and (**d**), respectively). SEM images of $SmCo_5/20\%Fe$ annealed at 600 °C for 1 h (figures (**e**) and (**f**)). A 50 μm scale was used for (**a**), (**c**) and (**e**), and a 5 μm scale was used for (**b**), (**d**) and (**f**), with an electron energy of 15 keV and magnification of ×500 (**a,c,d**) and ×5000 (**d–f**).

The composition of the samples was investigated by EDX in the SEM. The net composition of all samples corresponds with that of $SmCo_5/20\%Fe$. However, the amount of CaO could not be evaluated. Whereas 5% was added initially, some of the oxide had adhered to the mixing vial of the Turbula mixer; therefore, it is likely that the actual content of CaO inside the samples is significantly less. Furthermore, considering that O is not well-detected by the EDX method, the amount of CaO retained in the sample could not be determined.

The degree of interphase exchange coupling in the produced magnetic nanocomposites was investigated using demagnetization curves at low temperature (4 K). The reasoning behind this approach is that at cryogenic temperatures, the anisotropy of the $SmCo_5$ phase is

higher. Moreover, this also means that the condition for interphase exchange is also stricter (i.e., the domain wall width for $SmCo_5$ should be smaller at low temperature) [26,27].

The magnetic measurements at low temperature (Figure 3) show that all samples produced with CaO addition have excellent magnetic properties and high coercivity (approximately 1.8 T for all samples); the demagnetization curves are smooth and present no visible kinks, which is indicative of good interphase exchange coupling. On the other hand, the $SmCo_5$/Fe nanocomposites present lower coercivity. The $SmCo_5$/Fe sample annealed for 1 h presents very poor coercivity (0.74 T)—approximately half that of the $SmCo_5$/20%Fe + 5%CaO sample annealed for the same duration. This observation further strengthens our previous argument, that the addition of CaO inhibits Fe diffusion, with the larger Fe crystallites in the $SmCo_5$/Fe sample leading to a decrease in the effectiveness of the interphase exchange. These facts, together with the higher fraction of the 2:17-type phase, lead to a stark drop in coercivity. CaO does not altogether prevent Fe diffusion, as evidenced by the reduction in remanence when the annealing time is increased from 0.5 h to 1 h. The effect of washing away the CaO is also visible in Figure 3a. Whereas the washed sample presents nearly the same coercive filed $H_c$ as before washing, the remanence drops where an increase was expected. The drop is likely due to some superficial oxidation during the washing process (values are given in Table 2). It is very likely that the method used to remove oxygen and water from the solvents was not completely effective; however, with an improved solution, the oxidation may be reduced in the future.

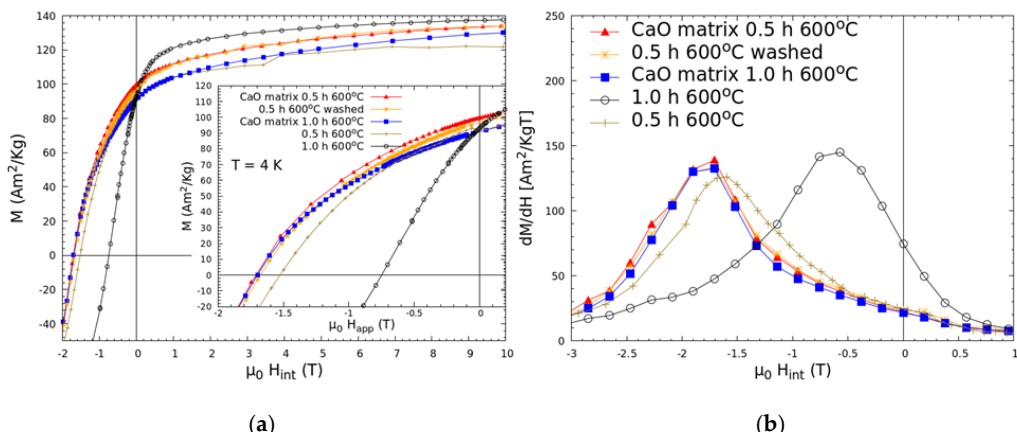

| (a) | (b) |

**Figure 3.** Demagnetization curves (**a**) and *dM/dH* vs. *H* plots (**b**) recorded at 4 K for $SmCo_5$/20%Fe nanocomposites annealed at 600 °C with and without CaO addition. The data for $SmCo_5$/20%Fe annealed for 0.5 h are taken from [21].

**Table 2.** Summary of magnetic characteristics for $SmCo_5$/20%Fe nanocomposites produced with and without CaO and annealed at 600 °C for up to 1 h. The data for the $SmCo_5$/Fe sample annealed for half an hour are taken from references [21].

| | **4 K** | | | | | **300 K** | | | | |
|---|---|---|---|---|---|---|---|---|---|---|
| **Sample** | $H_c$ | $M_r$ | $M_s$ | $M_r/Ms$ | $(BH)_{max}$ | $H_c$ | $M_r$ | $M_s$ | $M_r/Ms$ | $(BH)_{max}$ |
| $SmCo_5$/20%Fe | T | $Am^2/Kg$ | | - | $kJ/m^3$ | T | $Am^2/Kg$ | | - | $kJ/m^3$ |
| 600 °C 0.5 h | 1.52 | 102 | $125 \pm 4$ | 0.82 | 145 | 0.83 | 91 | $127 \pm 4$ | 0.72 | 135 |
| 600 °C 1 h | 0.74 | 93 | $135 \pm 4$ | 0.69 | 90 | 0.35 | 76 | $135 \pm 4$ | 0.56 | 45 |
| +CaO 600 °C 0.5 h | 1.71 | 100 | $131 \pm 4$ | 0.76 | 173 | 0.97 | 94 | $127 \pm 4$ | 0.74 | 136 |
| +CaO 600 °C 1 h | 1.70 | 91 | $122 \pm 4$ | 0.75 | 146 | 0.96 | 84 | $118 \pm 4$ | 0.71 | 112 |
| +CaO 600 °C 0.5 h washed | 1.67 | 97 | $128 \pm 4$ | 0.76 | 162 | - | - | - | - | - |

In order to better evaluate the degree of interphase exchange coupling, *dM/dH* vs. *H* plots were also calculated (Figure 3b). These plots show a single maximum at high

magnetic field values for all of the samples produced with CaO addition. This maximum is centered around $H_c$. In the case of the SmCo$_5$/Fe nanocomposite sample without CaO addition, an intense maximum is observed at a slightly lower field when annealed for 0.5 h. Moreover, to the right-hand side of the maximum, the curve has a lower slope compared to the samples made with CaO addition. This difference in slope could be attributed to the existence of a larger number of ineffectively coupled soft-phase grains. Moreover, when the annealing time is increased to 1 h, for the SmCo$_5$/Fe sample, the maximum is centered around $-0.7$ T, and a small shoulder is visible at high field between $-2$ and $-2.5$ T, which likely corresponds to a poorly coupled hard magnetic material, SmCo$_5$.

The hysteresis loops at room temperature (300 K) are presented in Figure 4a. The significant disparity between the samples made with and without CaO remains at higher temperatures, only now, the gap has widened for the samples annealed for 1 h. The coercivity of the samples produced using CaO is nearly three times that of the SmCo$_5$/Fe nanocomposite. The *dM/dH* vs. *H* curves show the most variation with increased temperature. All of the peaks become sharper and are centered around the new $H_c$ values. The reduction in the broadness of the peaks can be related to the improved degree of interphase exchange coupling due to the more forgiving critical dimensions. The increased sharpness and intensity of the peaks also signify that more reversal processes are concentrated around the values of the coercive fields at room temperature as compared to low temperature, where the peaks are very broad, indicating a less homogeneous reversal. An improved interphase exchange can also be observed in the SmCo$_5$/Fe samples. The slopes of the peaks increase, and for the sample annealed for 1 h, the shoulder at high field disappears, i.e. the decoupled hard-phase peak observed at 4 K is now coupled at 300 K and reverses at lower fields with the rest of the material due to the decrease in anisotropy with temperature. Furthermore, a difference in the peak intensity between the SmCo$_5$/20%Fe + 5%CaO samples can be observed, with the lower values seen for the 1 h annealed sample likely due to the aforementioned superficial oxidation.

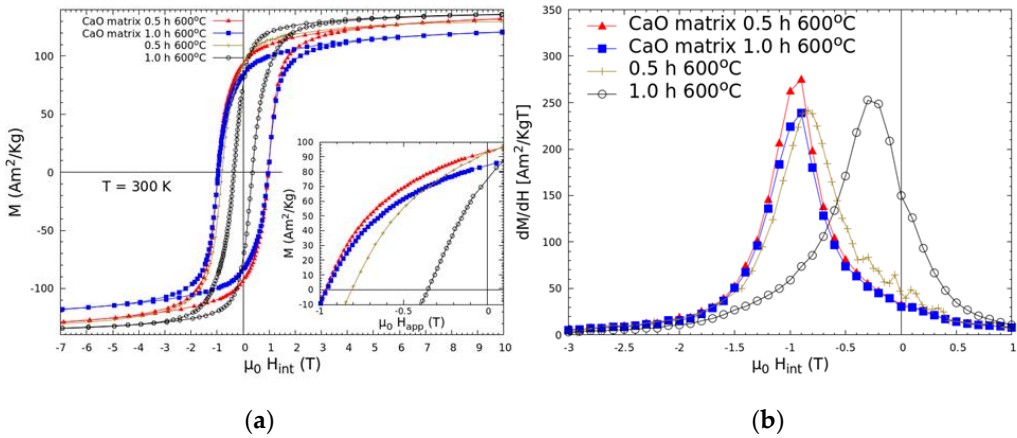

**Figure 4.** Hysteresis curves (**a**) and *dM/dH* vs. *H* plots (**b**) recorded at 300 K for SmCo$_5$/20%Fe nanocomposites annealed at 600 °C with and without CaO addition. The data for SmCo$_5$/20%Fe annealed for 0.5 h are taken from [22].

A summary of the most relevant magnetic properties is presented in Table 2. The literature reference sample presented for the virgin material annealed for 0.5 h proves that even with shorter annealing times, CaO addition is beneficial for interphase exchange coupling, yielding a 10% increase in $H_c$; however, with a similar cost in terms of $M_r$. Moreover, even with a lower remanence, the energy product improves at 4 K from 145 kJ/m$^3$ to 173 kJ/m$^3$ with CaO addition. Furthermore, whereas the energy product for the sample without CaO addition diminishes by 38 % with an additional 0.5 h of annealing, the sample produced with CaO only diminishes by 16% under the same conditions. However, at room temperature, the differences between the samples annealed for 0.5 h diminish, with two

samples producing similar energy products, i.e., 135 kJ/m$^3$. On the other hand, when the annealing time is increased to 1 h, the sample produced using CaO maintains three times the energy product compared to the SmCo$_5$/Fe nanocomposite produced without CaO material.

### 3. Materials and Methods

The starting powders were SmCo$_5$ (production supply; MAGNETI Ljubljana d.d., Slovenia, jet-milled particles <40 μm), Fe (produced by inert gas atomization, size <40 μm; Högnäs AB, Luleå, Sweden) and CaO (Reagent grade, Alfa Aesar, Kandel, Germany).

Two sample powders were produced: (i) SmCo$_5$/20%Fe, consisting of a 4-to-1 weight ratio of SmCo$_5$ to Fe, and (ii) (SmCo$_5$/20%Fe) + 5%CaO, consisting of the same 4-to-1 weight ratio mixture of SmCo$_5$ to Fe, to which another 5 wt % CaO was added. The CaO was placed in a furnace at 100 °C beforehand in order to eliminate any absorbed water. The starting powders were mixed for 30 minutes using a Turbula mixer. The mixed powders for each sample were mechanically milled for 6 hours in a Fritsch (Idar-Oberstein, Germany) Pulverisette 4 planetary ball mill. The milling vials used were 80 mL 440C stainless steel vials. In each vial, 26 steel balls with a diameter of 10 mm were used. The ball-to-powder weight ratio was 10:1. The ratio between the vial and main disk rpm was −900/333.

The milled samples were then placed in Ta sample holders and sealed in a quartz tube with a vacuum valve. The tube was connected to a turbo-molecular pump and kept at high vacuum (10$^{-6}$ mbar). The milled nanocomposite samples were annealed at 600 °C for two durations: 0.5 h and 1 h. At the start of the annealing process, the furnace was pre-heated to the annealing temperature, and the evacuated quartz tube was inserted. Cooling was performed in the furnace.

After annealing, for one of the samples, the CaO was removed by washing. A 45 mL mixture of 1:1 by volume glycerin and ethanol was used as a solvent. The powder was sealed in a 50 mL eprouvette with the solvent and vigorously agitated by hand, followed by 10 min of ultrasonication in an ultrasonic bath. Agitation and sonication were repeated 3 times; then, the particles were separated from the solvent using a centrifuge. After separation, the solvent was refreshed, and the whole process was repeated 3 more times. At the end, the glycerin was cleaned by the same process. However, for this last step, only ethanol was used as a solvent. Finally, the powder was separated from the solution and left to dry. All of the solvents were degassed as well as possible before use by simultaneous heating, ultrasonication and evacuation (dynamical primary vacuum pump).

The structure and microstructure of the milled samples were investigated by X-ray diffraction (XRD) using a Bruker (Karlsruhe, Germany) D8Advance diffractometer equipped with a Cu source. The composition of the samples was investigated by the Rietveld method using the Fullprof program [17]. The crystallite sizes were estimated using the Scherrer method [18] from the full width at half maximum obtained by fitting the Fe peaks with two convoluted pseudo-Voigt functions, one for Cu K$\alpha_1$ and one for the Cu K$\alpha_2$ wavelength, separated by the corresponding spectral separation. The function corresponding to the K$\alpha_2$ wavelength is identical in shape to that corresponding to the K$\alpha_1$ wavelength but only half the intensity.

The microstructure and particles sizes of the powders were investigated by electron microscopy using a Jeol-JSM 5600 LV scanning electron microscope (SEM) (Jeol, Pleasanton, USA). This also allowed for the investigation of the chemical composition of the samples, as the microscope was equipped with an UltimMAX65 energy dispersive X-ray spectrometer (EDX) (Oxford Instruments, Bristol, UK).

The magnetic properties of the samples were investigated using a vibrating sample magnetometer produced by Cryogenics (Cryogenics, London, UK). For magnetic measurements, the samples were blocked in epoxy resin. Demagnetization curves were recorded at 4 K, and full hysteresis curves were recorded at 300 K in applied fields of up to 10 T. In order to produce more accurate results, pressed powder was also measured for each sample at 10 T, and the hysteresis curves were normalized to the saturation magnetization

of the powders. The internal magnetic field was calculated by approximation of dilute spherical particles [19]. The saturation magnetization ($M_s$) for the studied nanocomposites was determined using the saturation law approach [20]:

$$M(H) = M_s \left( 1 - \frac{a_1}{H} - \frac{a_2}{H^2} \right) + \chi H \tag{1}$$

where $H$ is the value of the magnetic field; $a_1$ and $a_2$ are coefficients that describe the low- and high-field part of the magnetization curve, respectively; and $\chi$ is the paramagnetic-like factor at high field.

All powder handling was performed under a pure inert Ar gas atmosphere inside of a glovebox (MBraun, Garching bei München, Germany). XRD and loading of samples into the SEM were performed in air.

### 4. Conclusions

Exchange-coupled nanocomposite $SmCo_5/20\%Fe$ with added 5 wt% CaO powders were successfully produced by mechanical milling and subsequent annealing. The addition of CaO has been shown to inhibit soft-phase grain growth and possibly the formation of $Sm_2Co_{17}$-type phase impurities.

The magnetic properties of the nanocomposites have been shown to improve when CaO is added, with an energy product improvement (at 4 K) of 20% in comparison to identically produced $SmCo_5/Fe$ samples (without CaO). Moreover, the addition of CaO helps to maintain the microstructure and magnetic properties for longer at high temperatures. When increasing annealing time from 0.5 h to 1 h at 600 °C, the $(BH)_{max}$ of the material without CaO decreased by two-thirds, whereas the material produced with CaO addition only lost 17%.

In conclusion, CaO addition to $SmCo_5/20\%Fe$ is beneficial for the magnetic properties of the nanocomposite obtained by mechanical milling. CaO addition should be beneficial for consolidation techniques, such as spark plasma sintering, as the additive makes the microstructure more resilient at the high temperatures involved in the consolidation process.

**Author Contributions:** Funding acquisition, R.H.; Investigation, R.H., R.A.B., F.P. and G.S.; Project administration, R.H. and V.P.; Supervision, V.P. and O.I.; Writing-original draft, review and editing, R.H., O.I. and V.P. All authors have read and agreed to the published version of the manuscript.

**Funding:** This work was made possible through the financial support of the Romanian Ministry of Research, Innovation and Digitalization (grant PN-III-P2- 2.1-PED-2019-4696).

**Data Availability Statement:** The data presented in this study are available on request from the corresponding author.

**Conflicts of Interest:** The authors declare no conflict of interest.

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
