# Peer review of "Structural, Microstructural and Magnetic Properties of SmCo5/20wt%Fe Magnetic Nanocomposites Produced by Mechanical Milling in the Presence of CaO"

_magnetochemistry, doi:10.3390/magnetochemistry8100124_

Round 1

Reviewer 1 Report

The manuscript “structural, microstructural and magnetic properties of SmCo5/20wt%Fe magnetic nanocomposite produce by mechanical milling in the presence of CaO” by Hirian et al. was reviewed. The authors have reported the introduction of CaO as an additive precursor for the inhibition of grain growth of the soft magnetic phase during annealing via the mechanical milling synthetic route. The manuscript is of significant interest and contribution to the scientific field of knowledge. The investigation and control of phase purity is of critical importance in nanotechnology and its applicability. The authors have also demonstrated good knowledge of the subject and presented a logical discussion. The manuscript may be considered for publication after the following updates have been reflected.

1.     Hc and (BH)max should be defined in the abstract

2.     The abstract section should be strengthen with the benefits/advantages of phase stability in the field application of magnetic nanocomposites

3.     Lines 61 and 64, a suitable references should be supplied to support the claims

4.     Please supply TEM/HRTEM images of the nanocomposites to enrich and enhance discussions

5.     Possibly characterizations such as lattice fringes, SAED can be utilize to examine phase and crystallinity of nanocomposites

6.     Possibly do a particle size distribution of the particle grain sizes recorded

7.     The references are quite old, only 4 out of the 22 references can be regarded as recent. 

Reviewer 2 Report

In this work the authors present a careful study regarding the effects of using CaO as an additive in the preparation of SmCo5/Fe nanocomposites. The results are reported from 4 sets of samples milled for the same time with and without the addition of CaO and post annealed for either 30 or 60 minutes. Part of the results are also reported fort the sample milled with CaO and annealed for 30 minutes after being subjected to a procedure to wash away CaO residues. From X ray diffraction results the authors propose that the CaO additive limits Fe diffusion and the growth of the reportedly detrimental Sm­2Co17 phase. Scanning electron microscopy imaging revealed that samples milled with the additive formed micrometer-sized particulates while, in absence of CaO, the material formed larger flakes and more compact aggregates. The reported EDX analysis was comparable to SmCo5/20%Fe composite phase for all samples. Through magnetization characterizations at 4K and 300K the authors finally report slightly superior properties of the samples prepared with the additive in terms of coercivity, magnetization of remanence, magnetization of saturation, and the product “BH”. Although of limited interest and originality, the manuscript is clearly written, the results are objectively reported and support the main hypothesis proposed by the authors. For these reasons I recommend the publication of this work on the special issue of Magnetochemistry, given the following 3 minor concerns are addressed.

1 - Throughout the introduction the authors refer to the “quality” of the nanocomposite based on the size of the soft magnet crystallite as a limiting factor regarding the preservation of its exchange-induced stiffness. However, the last sentence also associates the grain size to the thermal stability of the nanocomposite. The authors do not elaborate on this second “quality” parameter. I would like to kindly request the authors to better explain this association or add a reference which makes this relation clearer.

2 - Could the authors comment, even qualitatively, on why the possibility of iron oxidation arising from introduction of oxygen through the addition of CaO does not seem to be an issue? I would also recommend to emphasize in the manuscript that, even after annealing, there was no evidence of iron oxide peaks on the diffractograms.

3 – Please, add references regarding the statements on the anisotropy of SmCo5 (line 134) and Sm2Co17 (line 136).

Reviewer 3 Report

Authors in the manuscript “Structural, Microstructural and Magnetic Properties of SmCo5/20wt%Fe Magnetic Nanocomposites Produced by Mechanical Milling on the Presence of CaO” report on the preparation and characterization of the novel exchange coupled hard–soft magnetic nanocomposites. In my opinion the topic is up-to-date. The studies of such novel functional nanomaterials are very attractive.

Authors revealed that CaO addition to SmCo5/20%Fe improves magnetic properties of the obtained nanocomposities.

It is a pity that the authors only used the XRD technique for structural studies. The structural characterization of nanomaterials is quite difficult task. The standard XRD diffraction technique is not sensitive enough to detect traces of nanophases. Moreover, due to the high degree of convolution between investigated phases and low signal to ratio the phase composition could be only estimated. In my opinion more valuable information about Fe and Sm2(Fe/Co)17 phase could be obtained from the Mössbauer spectroscopy measurements.

Nevertheless, I have a question about the XRD results. It seems to me that the peaks observed for 2q between 44 and 46 deg are clearly shifted towards lower angles after CaO adding. In my opinion it is observed for both samples annealed for 0.5 h as well 1 h. Does that have any explanation?

In addition, there are minor linguistic errors. For example, on the first page, line 41 should read: size of inclusion.

In my opinion, the presented results are novel and can be published, however should be reviewed again after significant revision.

Round 2

Reviewer 3 Report

In my opinion the manuscript can be published in the present form.